# Molecular Evolution of CatSper in Mammals and Function of Sperm Hyperactivation in Gray Short-Tailed Opossum

**DOI:** 10.3390/cells10051047

**Published:** 2021-04-29

**Authors:** Jae Yeon Hwang, Jamie Maziarz, Günter P. Wagner, Jean-Ju Chung

**Affiliations:** 1Department of Cellular and Molecular Physiology, Yale School of Medicine, New Haven, CT 06510, USA; 2Department of Ecology and Evolutionary Biology, Yale University, New Haven, CT 06520, USA; jamie.maziarz@yale.edu (J.M.); gunter.wagner@yale.edu (G.P.W.); 3Yale Systems Biology Institute, Yale University, West Haven, CT 06516, USA; 4Department of Obstetrics, Gynecology and Reproductive Sciences, Yale School of Medicine, New Haven, CT 06510, USA

**Keywords:** marsupials, gray short-tailed opossum, sperm, hyperactivated motility, CatSper

## Abstract

Males have evolved species-specifical sperm morphology and swimming patterns to adapt to different fertilization environments. In eutherians, only a small fraction of the sperm overcome the diverse obstacles in the female reproductive tract and successfully migrate to the fertilizing site. Sperm arriving at the fertilizing site show hyperactivated motility, a unique motility pattern displaying asymmetric beating of sperm flagella with increased amplitude. This motility change is triggered by Ca^2+^ influx through the sperm-specific ion channel, CatSper. However, the current understanding of the CatSper function and its molecular regulation is limited in eutherians. Here, we report molecular evolution and conservation of the CatSper channel in the genome throughout eutherians and marsupials. Sequence analyses reveal that CatSper proteins are slowly evolved in marsupials. Using an American marsupial, gray short-tailed opossum (*Monodelphis domestica*), we demonstrate the expression of CatSper in testes and its function in hyperactivation and unpairing of sperm. We demonstrate that a conserved IQ-like motif in CatSperζ is required for CatSperζ interaction with the pH-tuned Ca^2+^ sensor, EFCAB9, for regulating CatSper activity. Recombinant opossum EFCAB9 can interact with mouse CatSperζ despite high sequence divergence of CatSperζ among CatSper subunits in therians. Our finding suggests that molecular characteristics and functions of CatSper are evolutionarily conserved in gray short-tailed opossum, unraveling the significance of sperm hyperactivation and fertilization in marsupials for the first time.

## 1. Introduction

To win the competition between male rivals over females and breed successfully [1], males have evolved unique reproductive strategies. The rapid evolution of sperm design, such as morphology, swimming patterns, and/or cell numbers in ejaculates, can establish the most successful strategy to fertilize the eggs [2]. Mammalian males have evolved increased sperm numbers in their ejaculate to overcome the physical, chemical, and anatomical obstacles in the female reproductive tract; acidic environment near the vagina, cervical mucus, fluid flow in the uterine and the oviduct, narrow path of the uterotubal junction (UTJ), and/or the immune systems limit the number of sperm cells to reach to the fertilizing sites [3,4].

A small number of mammalian sperm cells that arrive at the fertilizing site show unique motility patterns called hyperactivated motility, characterized by asymmetric tail beating with increased flagellar amplitude [5,6]. In mice, hyperactivated motility enables sperm to pass UTJ [7], to swim efficiently under the viscous fluid in the oviductal lumen [8], and to detach from the oviductal reservoir to approach the eggs [9]. In addition, hyperactivation is required to penetrate the glycoprotein barrier of the oocytes, the zona pellucida [10,11]. Currently, sperm hyperactivation is reported in eutherian and monotreme species [12] but not in non-mammalian vertebrates. These results suggest that sperm hyperactivation is a shared physiological event in mammalian fertilization. Yet, the extent to which hyperactivated motility is conserved and its regulation and function have diverged in mammal fertilization is still not well studied.

Hyperactivated motility is triggered by Ca^2+^ influx into sperm cells through the sperm-specific ion channel, CatSper—a multi-protein complex composed of four pore subunits (CatSper1, 2, 3, and 4) and accessory subunits (CatSperβ, γ, δ, ε, ζ, and EFCAB9) [13]. Previous studies demonstrated that genetic aberration of the transmembrane subunits such as CatSper1–4, δ and ε lead to male infertility in mice or humans due to the failure to develop hyperactivated motility [13]. Non-transmembrane subunits, EFCAB9 and CatSperζ, form a binary gate-keeper complex and regulate CatSper channel activity by sensing changes in the intracellular Ca^2+^ levels and pH [14,15]. This CatSper-mediated Ca^2+^ signaling orchestrates the downstream molecular events such as capacitation-associated tyrosine phosphorylation (pTyr) in sperm cells [7]. Our recent study also revealed that mouse sperm that successfully arrive at the ampulla-the fertilizing site- are recognized by intact CatSper1 subunit and a low degree of pTyr [16]. These findings suggest that the molecular and functional integrity of the CatSper channel might be associated with sperm selection mechanisms in mammals. CatSper subunit sequences are annotated in diverse animals [15,17]. However, the extent to which the molecular workings of the CatSper channel are evolutionarily conserved across mammals, especially in distantly related mammalian species, remains largely unexplored.

Here, we report molecular evolution and conservation of the CatSper channel in the genome throughout eutherians and marsupials (therian mammals). Using gray short-tailed opossum (*Monodelphis domestica*) as an animal model of marsupials, we explored the expression and physiological significance of the CatSper channel and hyperactivated motility in marsupials. CatSper subunits express in post-meiotic male germ cells in the opossum. Paired opossum sperm develop hyperactivated motility asynchronously within the two flagella. Successful hyperactivation of paired and unpaired opossum sperm requires Ca^2+^ influx, likely through CatSper. For the first time, we reveal that mouse and human CatSperζ proteins contain a conserved IQ-like motif, which is essential for its interaction with EFCAB9. We demonstrate that opossum EFCAB9 can complex with mouse CatSperζ in the IQ-like motif-dependent manner. Our findings suggest that molecular characteristics and physiological function of CatSper are evolutionarily conserved throughout the therian mammals.

## 2. Materials and Methods

### 2.1. Protein Sequence Comparison and Phylogenic Analysis

Protein sequences of the annotated CatSper orthologs from 73 therian mammals were from the NCBI gene database (Appendix A; https://www.ncbi.nlm.nih.gov/gene/, accessed on 30 March 2021). Protein sequences of the orthologs were aligned with MUSCLE [18] and concatenated with the following order; CatSper1-2-3-4-β-γ-δ-ε-ζ-EFCAB9 or CatSper1-2-3-4-β-γ-δ-ε--EFCAB9. Pairwise distances between the concatenated sequences were calculated using MEGA7 with the default option [19]. The calculated pairwise distances were represented as heatmaps or violin plots using GraphPad Prism 8.0 (GraphPad Software Inc., San Diego, CA, USA). Rooted and unrooted phylogenic trees were generated with the divergence time between species referred from TimeTree ([20]; http://www.timetree.org/, accessed on 30 March 2021) or with the concatenated CatSper sequences, respectively. The unrooted phylogenic trees were constructed by maximum-likelihood analysis with 500 bootstrap replications using MEGA7 [19].

### 2.2. Protein Sequence Alignment and IQ Motif

Protein sequences of human, mouse, and opossum EFCAB9 and human and mouse CatSperζ were aligned using Clustal Omega (https://www.ebi.ac.uk/Tools/msa/clustalo/, accessed on 20 March 2021). Candidate calmodulin (CaM) binding IQ-like motifs in human and mouse CatSperζ were searched using Calmodulin Target Database ([21]; http://calcium.uhnres.utoronto.ca/ctdb/ctdb/sequence.html, accessed on 24 March 2021).

### 2.3. Animals

Gray short-tailed opossums (*M. domestica*) and wild-type C57BL/6 mice were treated in accordance with the protocols approved by the Institutional Animal Care and Use Committee (#11313 and #11483 for gray short-tailed opossums; #20079 for mice) for Yale University. Juvenile (2 months old, *n* = 2) and adult (4–5 months old, *n* = 5; >18 months old, *n* = 4) male gray short-tailed opossums and adult male mice (>90 days old, *n* = 3) were used in this study.

### 2.4. Sperm Collection and Capacitation

Epididymal spermatozoa from adult gray short-tailed opossums (>4 months old) and mice (>90 days old) were collected by swim-out methods. Briefly, collected cauda epididymis was placed in M2 medium (MilliporeSigma, Burlington, MA, USA) at 37 °C for 10 min (mouse) or 20 min with gentle rocking (opossum). For capacitation, the collected opossum and mouse sperm were washed one time by centrifugation with M2 medium and incubated in human tubular fluid (HTF) medium (MilliporeSigma, Burlington, MA, USA) at 37 °C, 5% CO_2_ condition for 90 min. In order to induce sperm capacitation under Ca^2+^ chelated or CatSper channel inhibiting condition, 2.5 mM EGTA or 10 μM NNC 55-0369 (NNC, Alomone Labs, Jerusalem, Israel) were supplemented in HTF medium, respectively.

### 2.5. Flagella Waveform Analysis

Uncapacitated or capacitated opossum sperm were transferred to the imaging chamber for Delta-T culture dish controller (Bioptechs, Butler, PA, USA) containing 37 °C HEPES-buffered HTF medium with 0.5% methylcellulose (*w*/*v*) [14]. After 1 min, the flagellar movement of head-tethered sperm was recorded for 2 s with 200 fps speed using pco.edge sCMOS camera equipped in Axio observer Z1 microscope (Carl Zeiss, Oberkochen, Germany). FIJI software [22] was used to measure α-angle and beat frequency of the sperm flagella and to draw overlaid images of flagella for two beat cycles as described in a previous study [14].

### 2.6. Swimming Trajectory Analysis

To analyze swimming trajectory, opossum sperm cells were transferred to 37 °C H-HTF medium with 0.5% methylcellulose (*w*/*v*) in an imaging chamber coated with 0.2% agarose to minimize head attachment on the plate. Sperm movements were recorded for 2 s with 100 fps speed using Axio observer Z1 microscope with pco.edge sCMOS camera (Carl Zeiss, Oberkochen, Germany). Overlaid images to trace the sperm swimming paths were generated by using FIJI software.

### 2.7. Sperm Fluorescence Staining

Uncapacitated and capacitated short gray-tailed opossum sperm cells were washed with PBS and attached on the glass coverslip by centrifugation at 700× *g* for 5 min. The sperm cells were fixed with 4% PFA for 10 min at room temperature (RT) followed by washing with PBS three times. The fixed sperm were permeabilized with 0.1% Triton X-100 in PBS at RT for 10 min and blocked with 10% normal goat serum in PBS for an hour at RT. The opossum sperm were stained with 10 μg/mL of mouse monoclonal phosphotyrosine antibody (clone 4G10, MiliporeSigma, Burlington, MA, USA) or Alexa-568 conjugated peanut agglutinin (PNA, Invitrogen, Carlsbad, CA, USA) in blocking solution at 4 °C for overnight. Immunostained coverslips were washed with PBS three times and incubated with goat anti-mouse IgG conjugated with Alexa-568 (1:1000; Invitrogen, Carlsbad, CA, USA) in blocking solution for an hour at RT. Stained samples were mounted with Vectashield (Vector Laboratories, Burlingame, CA, USA) and imaged with Zeiss LSM710 Elyra P1 using Plan-Apochrombat 63X/1.40 oil objective lens. Hoechst (Invitrogen, Carlsbad, CA, USA) was used for counterstaining.

### 2.8. Scanning Electron Microscopy

Opossum sperm cells were attached on the glass coverslips and fixed with 2.5% glutaraldehyde (GA) in 0.1 M sodium cacodylate buffer (pH7.4) for one hour at 4 °C followed by post-fixation with 2% osmium tetroxide in 0.1 M cacodylate buffer. Post-fixed samples were washed with 0.1 M cacodylate buffer three times and dehydrated through a series of ethanol to 100%. Fixed samples were dried using a Leica 300 critical point dryer with liquid carbon dioxide. The coverslips were glued to aluminum stubs and sputter-coated with 5 nm platinum using a Cressington 208HR (Ted Pella Inc, Redding, CA, USA) rotary sputter coater. Samples were imaged with a Hitachi SU-70 scanning electron microscope (Hitachi High-Technologies, Tokyo, Japan).

### 2.9. RNA Extraction and qRT-PCR

Opossum testicular RNA was extracted from males aged 2, 4–5, and over 18-month-old using RNeasy mini-kit (Qiagen, Hilden, Germany), and 500 ng of the total RNA was synthesized to cDNA using iScript cDNA Synthesis Kits (BioRad, Hercules, CA, USA). The cDNA samples were subjected to qRT-PCR (CFX96, BioRad, Hercules, CA, USA) or cloning open reading frame (ORF) of opossum EFCAB9. Primer pairs used for the qRT-PCR were listed in Appendix A. *TBP* was used for the reference to normalize expression level, and relative mRNA expression levels of each CatSper subunit were calculated by the ddCt method.

### 2.10. Expression Constructs and Molecular Cloning

Mouse CatSperζ (*pCAG-mCatSperz-V5*) and EFCAB9 (*phCMV3-mEFCAB9-HA*) expression constructs were generated in previous study [14,15]. Mouse CatSperζ ORF was subcloned into phCMV3 to generate constructs expressing FLAG-tagged full-length (*phCMV3-mCatSperz-FLAG*) and N- or C-terminus truncated CatSperζ (*phCMV3-mCatSperz-∆N21-FLAG* or *phCMV3-mCatSperz-∆C21-FLAG*, respectively). To generate the construct expressing mouse CatSperζ with a mutation at IQ-like motif sequences (L174A/Q175A), mouse *CatSperz* cDNA was site-directed mutagenized and subcloned into phCMV3 (*phCMV3-CatSperz-AA-FLAG*). Human *CatSperz* and *EFCAB9* ORFs [15] were subcloned into phCMV3 (*phCMV3-hCatSperz-V5* and *phCMV3-hEFCAB9-HA*, respectively). Opossum EFCAB9 ORF was amplified from the testis cDNA library using NEB Q5 DNA Polymerase (NEB, Ipswich, MA, USA), and the PCR product was extracted for blunt-end cloning (ClonJET PCR Cloning Kit, ThermoFisher, Waltham, MA, USA). The opossum EFCAB9 ORF clone was sequenced and subcloned into phCMV3 vector (*phCMV3-OpEFCAB9-HA*). All subcloning into phCMV3 were carried out with NEBuilder^®^ HiFi DNA Assembly (NEB, Ipswich, MA, USA). For FLAG or V5 tags, a stop codon was placed at the upstream of HA-encoding sequences in the phCMV3 vector.

### 2.11. Transient Protein Expression in Mammalian Cells

Human embryonic kidney 293T cells (HEK293T; ATTC, Manassas, VA, USA) were cultured in DMEM (Gibco, Gaithersburg, MD, USA) supplemented with 10% FBS (ThermoFisher, Waltham, MA, USA) and 1X Pen/Strep (Gibco, Gaithersburg, MD, USA) at 37 °C, 5% CO_2_ condition. The cultured HEK293T cells (passage 30–35) were transfected to express target proteins transiently. Polyethylenimine (PEI) was used for transfection, and transfected HEK293T cells were used for the co-immunoprecipitation.

### 2.12. Protein Co-Immunoprecipitation and Immunoblotting

The transfected 293T cells were lysed with 1% Triton X-100 in PBS supplemented with 1X EDTA-free protease inhibitor cocktail (cOmplete Mini, Roche, Basel, Switzerland) by rocking at 4 °C for 1 h. Lysates were centrifuged at 18,000× *g* for 30 min at 4 °C, and the supernatant was used for immunoprecipitation. The solubilized proteins in the supernatant were incubated with Anti-V5 agarose Affinity Gel (clone V5-10) or Anti-FLAG Affinity Gel (clone M2) from MilliporeSigma (Burlington, MA, USA ) or Pierce™ Anti-HA Magnetic Beads (clone 2-2.2.14) from ThermoFisher (Waltham, MA, USA) at 4 °C for overnight. The affinity gels and beads were washed with 1% Triton X-100 in PBS three times, and the immunocomplexes were eluted with 2X LDS containing 50 mM dithiothreitol (DTT) followed by boiling at 75 °C for 10 min. Input and co-immunoprecipitation (coIP) samples were subjected to SDS-PAGE and immunoblotted with the rabbit polyclonal α-EFCAB9 (1 μg/mL; [15]), rabbit monoclonal α-DYKDDDDK (1:2000; clone D6W5B, Cell Signaling Technology, Danvers, MA, USA), and mouse monoclonal α-FLAG (1:2000; clone M2, MilliporeSigma, Burlington, MA, USA), α-HA (1:2000; clone 2-2.2.14, ThermoFisher, Waltham, MA, USA), and α-V5 conjugated with HRP (1:1000; ThermoFisher, Waltham, MA, USA). HRP-conjugated goat anti-mouse or anti-rabbit IgG (1:10,000; Jackson ImmunoResearch, West Grove, PA, USA) were used for secondary antibodies.

### 2.13. Statistical Analysis

Statistical analyses were performed with a one-way analysis of variance (ANOVA) with the Tukey post hoc test. Differences were considered significant at * *p* < 0.05; ** *p* < 0.01; *** *p* < 0.001.

## 3. Results

### 3.1. CatSper Components Are Conserved and Evolved Slowly in Marsupials

Marsupials and eutherians diverged from their common ancestor around 160 million years ago (MYA) (Figure 1A and Appendix A). To understand the molecular evolution of CatSper components in marsupials, we performed comparative amino acid sequence analysis of CatSper proteins from 71 therian mammals including Tasmanian devil (*Sarcophilus harrisii*), a marsupial in which all the reported CatSper subunits are annotated in the genome (Appendix A). Pairwise distance analyses of the concatenated sequence of all ten CatSper subunits (CatSper1-2-3-4-β-γ-δ-ε-ζ-EFCAB9) revealed that Tasmanian devil CatSper protein sequences are highly divergent in therian mammals, together with CatSper proteins in rodents (Figure 1B, left, and Appendix A). The sequence comparison of the Tasmanian devil CatSper subunits to those in the eutherian species also support that Tasmanian devil CatSper proteins are distinct from eutherian CatSper proteins (Figure 1C, left). Interestingly, phylogenetic analyses showed that Tasmanian devil CatSper proteins are clustered together with those of Laurasiatherians despite their distinctive protein sequences (Appendix A). These results indicate that the molecular evolutionary patterns of CatSper proteins are different from taxonomic classification, suggesting that CatSper components might have evolved more slowly in the Tasmanian devil than in rodents. Thus, we normalized the pairwise distances of CatSper proteins by divergence time between two species and compared the values within therian mammals (Figure 1B, right, and Appendix A). Indeed, normalized pairwise distances reveal that Tasmanian devil CatSper proteins are rather less divergent than those in rodents considering their divergence time. Interclade comparison also supports the slower evolution of the CatSper proteins in the Tasmanian devil (Figure 1C, right). All these results indicate that CatSper proteins have evolved rapidly in rodents but slowly in the Tasmanian devil; marsupial CatSper proteins might have conserved physiological functions and molecular characteristics inherited from common ancestors of therian mammals.

### 3.2. Gray Short-Tailed Opossum Sperm Develop Hyperactivated Motility after Incubating under Capacitating Conditions

The comparative sequence analyses raised the possibility that the physiological function and molecular workings of the CatSper channel in sperm might have been conserved in marsupials (Figure 1). Currently, CatSper subunits are annotated fully or partly in five marsupial species (Appendix A): four Australian marsupials - Tasmanian devil (*S. harrisii*), koala (*Phascolarctos cinereus*), common brushtail possum (*Trichosurus vulpecula*), and common wombat (*Vombatus ursinus*) - and one American marsupial - gray short-tailed opossum (*M. domestica*, Figure 2A). Pairwise sequence analyses showed that CatSper proteins in gray short-tailed opossum and the Australian marsupials are sequence homologues and evolutionarily closer than eutherian CatSper proteins (Appendix A). These results suggest that gray short-tailed opossum and Australian marsupials are likely to share the physiological function and molecular characteristics of CatSper proteins despite their early divergence time (82 MYA, Appendix A).

The CatSper channel is activated under capacitating conditions, which enable eutherian sperm to beat asymmetrically and develop hyperactivated motility. However, it is not known whether marsupial sperm develop hyperactivated motility by activating the CatSper channel during capacitation. Thus, we examined flagellar movement and swimming patterns of sperm from gray short-tailed opossum, an established laboratory marsupial, before and after incubating under capacitating conditions. Opossum sperm are in paired or unpaired forms in the epididymis [23]. The acrosome localizes at the dorsal area of the sperm head, where two sperm form a pairing (Figure 2B,C) [23]. Scanning electron microscopy clearly shows that opossum sperm flagellum is composed of midpiece and principal piece separated by the annulus (Figure 2D). In addition, the longitudinal columns are also observed in the principal piece, just like the compartmentalized flagella in other eutherian species.

In order to understand how capacitation induces changes in opossum sperm motility, we compared their flagella beating patterns and swimming trajectory before and after incubating under capacitating conditions (Figure 3 and Appendix A). Flagella waveform analyses revealed that opossum sperm flagella beat asymmetrically after inducing capacitation (Figure 3A–C and Appendix A). We compared the maximum angle of the primary curvature (α-angle, [24]) (Figure 3B) and found that inducing capacitation often led one of the two flagella in paired sperm to beat asymmetrically first, followed by increasing the amplitude of beating flagella in both paired and unpaired sperm eventually (Figure 3A). Notably, capacitated single sperm show the highest flagellar amplitude, slowing down the beating frequency (Figure 3C). These temporal changes in the motility patterns indicate that hyperactivated motility helps to separate the paired sperm cells into single sperm cells, which are further hyperactivated. It was previously reported that paired opossum sperm swim linearly, and single sperm swim in circles [25]. We placed opossum sperm in viscous media to mimic the luminal environment of the female reproductive tract and to better visualize the details of the fast-swimming opossum sperm. We observed that inducing capacitation does not change the linear swimming pattern in paired sperm (Figure 3D and Appendix A). By contrast, the radius of the circular swimming path in single sperm increases after inducing capacitation, similar to what is observed for capacitated mouse sperm swimming under viscous conditions [8]. These results suggest that the characteristics of sperm hyperactivation are conserved in the gray short-tailed opossum. Yet, it serves a dual role: to dissociate paired sperm and to enable unpaired sperm to swim efficiently in the female reproductive tract.

### 3.3. Ca^2+^ Influx Is Required to Develop Hyperactivated Motility in Opossum Sperm

In eutherians, hyperactivated motility is triggered by Ca^2+^ influx into sperm cells [11,26,27,28]. It has also become evident that Ca^2+^ signaling negatively regulates capacitation-associated protein tyrosine phosphorylation (pTyr) [7,16,29]. Incubating sperm from gray short-tailed opossum under capacitating conditions enabled them to develop hyperactivated motility as well as pTyr (Figure 3E).

To test whether the Ca^2+^ requirement for sperm hyperactivated motility is conserved in gray short-tailed opossum sperm, we analyzed flagellar movement and swimming patterns of opossum sperm under capacitating but Ca^2+^-chelating conditions (Figure 4A,B and Appendix A). Opossum sperm capacitated with calcium-free conditions only vibrate and fail to develop hyperactivated motility (Figure 4A, + EGTA, and Appendix A). The defective flagellar movement seems to affect the free swimming of opossum sperm; the capacitated sperm fail to swim forward in the viscous medium (Figure 4B and Appendix A). We tested how treatment with a CatSper channel inhibitor, NNC, alters motility in the capacitated opossum sperm (Figure 4A,B and Appendix A). Although opossum sperm beat their flagella in the presence of NNC, the flagellar amplitude became relatively smaller than those of capacitated sperm without the drug treatment (Figure 4B and Appendix A). The impaired flagellar movement indicates that the NNC-treated sperm failed to develop hyperactivated motility. In addition, NNC-treated sperm swim inefficiently in a viscous medium, such as sperm capacitated in Ca^2+^ chelated conditions (Figure 4B and Appendix A).

Next, we examined the extent to which Ca^2+^ signaling is associated with pTyr development in opossum sperm during capacitation (Figure 4C,D). Both paired and single opossum sperm develop capacitation-associated pTyr. Intriguingly, opossum sperm develop pTyr only marginally during capacitation with EGTA, which is contrary to much more potentiated pTyr in mouse sperm capacitated under Ca^2+^-chelated conditions (Appendix A). Capacitation-associated pTyr developed similarly with or without NNC in both opossum and mouse sperm cells. These results elucidate that requirement of Ca^2+^ influx to develop hyperactivated motility is conserved in opossum sperm, likely mediated by the CatSper channel. Yet, pTyr development is not tightly linked to the Ca^2+^-signaling, which might have been evolved differently from eutherian to gray short-tailed opossum sperm cells.

### 3.4. Interaction of CatSperζ and EFCAB9 Is Conserved in Therian Mammals

In the presence of a CatSper channel inhibitor, the ability of gray short-tailed opossum sperm to develop hyperactivated motility is compromised (Figure 4). These results suggest that CatSper-mediated Ca^2+^ signaling, as it does in eutherian mammals, triggers sperm hyperactivation in the opossum. To test the hypothesis that the CatSper channel is functionally expressed in the opossum, we first examined mRNA expression of CatSper subunits in opossum testis (Figure 5). We were able to detect mRNA expression of all CatSper subunits with the exception of *CatSperz,* which is not annotated currently (Appendix A). CatSper pore subunits (*CatSper1*, *2*, *3*, and *4)* and *EFCAB9* express higher in adult (over 4–5 months old) testes than those in juvenile (2 months old) testes (Figure 5A). By contrast, *CatSperg*, *d*, and *e* expression were readily detected in juvenile testes while still lower than those in adult testes. These overall post-meiotic expression of CatSper pore subunits and EFCAB9 genes and the expression of other auxiliary subunits ahead of them in developing testes were previously reported [14,15,30], indicating conserved CatSper subunit expression patterns in the gray short-tailed opossum.

EFCAB9 and CatSperζ form a binary complex that is responsible in part for sensing intracellular pH and Ca^2+^ to regulate CatSper channel activity [15]. Mouse and human recombinant EFCAB9 and CatSperζ can interact with each other across species despite sequence variability of the eutherian CatSperζ orthologs (Figure 5B,C) [14,15]. CatSperζ is conserved only in mammals [14,15]. Thus, the EFCAB9-CatSperζ complex is a new molecular design specific to placental mammals in regulating CatSper channel activity. Currently, marsupial *CatSperz* orthologs are annotated in Tasmanian devil, koala, and common wombat (Appendix A). Genomic loci encoding CatSperζ and its neighboring genes, however, are conserved synteny in therians (Appendix A). These genomic characteristics suggest CatSperζ is likely to be present in the gray short-tailed opossum genome but not yet annotated, likely due to its sequence variability. Contrary to CatSperζ, its binding partner, EFCAB9, is conserved broadly in animals [15], and the amino acid sequence of opossum EFCAB9 is well aligned with those of human and mouse EFCAB9 (Figure 5D). As EFCAB9 interacts with CatSperζ across eutherian species (Figure 5C) despite sequence variability of CatSperζ orthologs, we hypothesized that opossum EFCAB9 could form a complex with eutherian CatSperζ if the complex formation is conserved in the opossum. Thus, we tested whether opossum EFCAB9 can interact with eutherian CatSperζ orthologs. We expressed opossum EFCAB9 transiently together with human or mouse CatSperζ in 293T cells and performed coIP (Figure 5E). Opossum EFCAB9 was found in the same complex with human and mouse CatSperζ, indicating that opossum EFCAB9 can interact with eutherian CatSperζ. These results suggest that CatSperζ-EFCAB9 complex formation is conserved in marsupials; the molecular mechanisms of pH-dependent Ca^2+^ sensing of the CatSper channel are likely shared among therian mammals.

### 3.5. EFCAB9 Signals via CatSperζ, the IQ-Like Motif Protein Specific to Therian Mammals

EFCAB9 is the sperm-specific CaM-like protein containing its unique, three EF-hand Ca^2+^ binding domains (Figure 5D; [15]). We searched whether CatSperζ orthologs contain CaM binding sites [21], such as IQ motif, as proteins containing the motif are well-known regulators of CaM. The search predicted that both human and mouse CatSperζ have an IQ-like motif at their C-termini (Figure 6A,B). To test whether the IQ-like motif is essential for the EFCAB9-CatSperζ interaction in therian mammals, we generated constructs encoding mouse CatSperζ with 21 amino acids deletion at N- (CatSperζ-∆N21) or C- (CatSperζ-∆C21) terminus (Figure 6C). The truncated mouse CatSperζ and EFCAB9 were heterologously expressed and subjected to coIP. Truncation of C-terminus containing the IQ-like motif, but not N-terminal deletion, severely impairs CatSperζ binding to EFCAB9 (Figure 6B,D). To further clarify the requirement of the IQ-like motif in CatSperζ and EFCAB9 interaction, we generated a construct encoding CatSperζ of which LQ are substituted to alanine (AA, CatSperζ-AA). This substitution also compromised CatSperζ and EFCAB9 interaction (Figure 6E). The altered interactions clearly demonstrate that IQ-like motif is essential for CatSperζ to form a binary complex with EFCAB9. Next, we tested whether the binding of opossum EFCAB9 to CatSperζ also relies on the IQ-like motif (Figure 6F,G). Opossum EFCAB9 cannot interact with mouse C-terminal truncated CatSperζ (Figure 6F) and the AA mutant (Figure 6G). The impaired interaction between opossum EFCAB9 and mouse CatSperζ-AA suggests that opossum CatSperζ also mediates EFCAB9 signaling via its IQ-like motif. Therefore, the IQ-like motif in CatSperζ is necessary to form EFCAB9 and CatSperζ binary complex, and the motif-dependent molecular interaction is conserved in therian mammals.

## 4. Discussion

### 4.1. Uniquely Paired Gray Short-Tailed Opossum Sperm Unpair during Capacitation

Sperm from American marsupials, including the gray short-tailed opossum, are paired during epididymal maturation (Figure 2; [31,32]), which is distinct from the unpaired epididymal sperm in Australian marsupials and eutherians. The paired sperm of Virginia opossum (*Didelphis virginiana*) can pass UTJ and maintain their physical interaction until the peri-fertilization period in the oviduct [33]. The pairing might contribute to efficient sperm migration to the fertilizing site in American marsupials. Motility analyses of gray short-tailed opossum sperm support this possibility; paired sperm swim faster than unpaired sperm in viscous conditions (Figure 3; [25]), suggesting that pairing can be a sperm cooperation mechanism. Sperm conglomerate in the epididymis is also known from egg-laying mammals, monotremes [34]. Similar to the pairing of epididymal sperm in American marsupials, platypus (*Ornithorhynchus anatinus*) and echidna (*Tachyglossus aculeatus*) sperm assemble into bundles of approximately 100 sperm cells in the epididymis [12,35]. Thus, the sperm interaction during epididymal maturation is an ancient trait from ancestor mammals to monotremes and American marsupials, but not to Australian marsupials and eutherians. In monotremes, an epididymal protein, SPARC, was detected from epididymal and ejaculated sperm bundle but not from the dissociated single sperm cells [12], suggesting SPARC is associated with sperm bundle formation. Yet, the molecular events associated with the sperm pairing in American marsupial species are unknown. Considering the conserved sperm conglomerate during epididymis transition in monotremes, SPARC or other epididymal proteins might be also involved in the sperm pairing of the gray short-tailed opossum.

Incubation under capacitating conditions in vitro unpairs the epididymal sperm of gray short-tailed opossum [25]. Previous studies observed unpaired or unpairing sperm cells from crypts of oviductal epithelium in Virginia opossum [33,36]. In the sperm undergoing unpairing, the peripheral junction is broken down, and the acrosome contents are revealed [33], enabling single sperm cells to interact with the zona pellucida efficiently. Therefore, capacitation-associated sperm unpairing in the oviduct must precede normal fertilization in American marsupials. Gray short-tailed opossum sperm develop hyperactivated motility after incubation under capacitating conditions (Figure 3). Hyperactivation alters the synchronized and symmetric flagellar beating of paired sperm cells, which would generate more force and facilitate dissociation. Intriguingly, under the capacitating conditions but lacking free Ca^2+^, we observed that two sperm were only weakly bound, and their pairing junction was almost displaced without developing hyperactivated motility (Figure 4). These results suggest that hyperactivated motility dissociates paired sperm mechanically, while additional molecular mechanisms are also involved in sperm unpairing during capacitation. In line with this idea, a previous study showed that increasing intracellular cAMP enhanced the dissociation of echidna sperm cells from the bundle [12].

### 4.2. Capacitation-Associated Signaling Events Are Distinct in Gray Short-Tailed Opossum from Eutherian Mammals

Inducing capacitation develops global pTyr in sperm cells by activating the PKA signaling pathway in eutherian species [37,38,39,40,41]. In our study, we found gray short-tailed opossum sperm also develop capacitation-associated pTyr (Figure 3 and Figure 4) similar to the report on the sperm of two Australian marsupials, the tammar wallaby (*Macropus eugenii*) and common brushtail possum (*T. vulpecula*) [42]. By contrast, echidna sperm cells do not develop pTyr when capacitated in vitro [12]. The absence of pTyr development in echidna sperm suggests that capacitation-mediated pTyr is an innovation in therian mammals. Although the physiological significance of pTyr in sperm remains unclear, previous studies have demonstrated that pTyr development is robustly enhanced in mouse sperm through genetic and pharmacological abolition of Ca^2+^ entry during capacitation [7,29]. In addition, a high concentration of Mg^2+^ further enhances pTyr development in mouse sperm capacitated in vitro in the absence of free Ca^2+^ [7]. This regulation elucidates that Mg^2+^ could be another player in regulating pTyr development. In gray short-tailed opossum sperm, however, pharmacological blockage of Ca^2+^ entry rather suppresses capacitation-associated pTyr development (Figure 4). We used HTF medium to capacitate opossum sperm as the overall composition of the tubular fluid of the female tract is shared in eutherian species [43]; bicarbonate, albumin, Ca^2+^, and nutrients are considered to be essential for sperm capacitation. Successful IVF was previously reported using a similar medium to HTF in gray short-tailed opossum [44]. Therefore, the observed difference is likely from different sensitivity of gray short-tailed opossum CatSper to Ca^2+^ blockage in pTyr development. Yet, it would be interesting to see whether the ion composition in the luminal fluid in the female tract and/or ion selectivity and sensitivity of CatSper is uniquely different in gray short-tailed opossum or other marsupials from in mouse and human. A recent mouse study observed that the small number of in vivo capacitated mouse sperm that arrive at the fertilizing site are mostly devoid of pTyr, while suppression of the inhibition was observed down along the female reproductive tract [16]. Thus, global phosphorylation might be a marker for degenerating sperm cells which should be eliminated from the female reproductive tract after fertilization occurs. Intriguingly, American marsupials produce around 10–100 times fewer sperm cells compared to Australian marsupials (except Dasyuridae species), yet 50% of the sperm cells can arrive at the upper oviduct 12 h after coitus [45]. Thus, the pTyr is not associated with sperm selection in gray short-tailed opossum sperm.

### 4.3. CatSperζ Conveys the IQ Domain Interaction with EFCAB9 and Species-Specific pH-Sensitivity in Therian Mammals

The CatSper channel is activated by intracellular alkalization in mice and humans [46,47]. Yet, the extent to which intracellular alkalization contributes to CatSper activation was shown to have species specificity. For example, increasing intracellular pH activates the CatSper channel in mouse and human sperm [46,48,49], but human sperm requires additional ligands for full activation, such as progesterone and/or prostaglandin E_1_. In mouse sperm, CatSper activation is further enhanced by increased intracellular Ca^2+^ concentrations [15]. A CaM-like molecule, EFCAB9, forms a complex with CatSperζ and senses Ca^2+^ for this modulation in a pH-dependent manner [15]. EFCAB9 is tightly bound to CatSperζ when intracellular pH is low, interacting with the channel pore and limiting Ca^2+^ influx before capacitation. The capacitation-induced increase in intracellular pH weakens the interaction and potentiates Ca^2+^ influx. We demonstrate that CatSperζ orthologs contain a conserved IQ-like motif, providing insights into CatSper regulation. Despite a high degree of sequence variability among CatSperζ orthologs [14,15], recombinant mouse CatSperζ can interact with opossum EFCAB9 via its IQ-like motif (Figure 5 and Figure 6), suggesting that a conserved IQ-like motif in CatSperζ orthologs serves as a binding site for EFCAB9 orthologs. In addition, the presence of IQ-like motifs in the CatSperζ orthologs from Tasmanian devil supports the conserved motif-mediated interaction between EFCAB9 and CatSperζ in mammals. Therefore, as seen in many other ion channels, Ca^2+^/CaM regulation via interaction with an IQ domain is a unified mechanism of CatSper regulation in therian mammals, yet the varying sequences of CatSperζ might convey different pH sensitivity in activating CatSper within therian mammals, including gray short-tailed opossum.

### 4.4. Regulation and Function of Sperm Hyperactivation has Evolved Distinctively in Mammalian Linages

In eutherians, Ca^2+^ enters sperm cells via CatSper and triggers hyperactivated motility [13]. The CatSper gene expression and physiological changes shared by eutherian mammals and gray short-tailed opossum (Figure 3, Figure 4 and Figure 5) highlight that CatSper-mediated Ca^2+^ signaling is an evolutionarily conserved molecular mechanism to trigger hyperactivated motility in therian species. It was demonstrated that hyperactivation is required for mouse, hamster, and human sperm to efficiently swim in the viscous media [50,51]. Sperm hyperactivation in mouse is also essential for migration in the female reproductive tract, especially passing the utero-tubal junction (UTJ) [16,30]. Thus, hyperactivation functions at multiple levels in eutherians: limiting sperm number in the oviduct, helping the sperm past the UTJ to navigate through the mucoidal oviduct, and finally penetrating the egg coat. The marsupial oviductal epithelium also secrets mucus, making the oviductal lumen viscous [52,53]. Yet, marsupial UTJ present few barriers to control sperm number [54]. This anatomical difference in the female reproductive tract suggests sperm hyperactivation does not play a major role in the sperm transition from the uterus to the oviduct in the gray short-tailed opossum. Accordingly, as many as around half of the inseminated sperm can pass the UTJ in American marsupials [45]. As both paired and unpaired opossum sperm are present in crypts of the oviduct [33,36], enhanced swimming ability of unpaired and hyperactivated opossum sperm under viscous fluid (Figure 3 and Figure 4) illuminates the functional conservation of sperm hyperactivation in the efficient navigation through the convoluted and viscous oviductal environment in eutherians and marsupials.

Unlike therian species, dissociated echidna sperm show asymmetric tail beating only when subjected to the chicken perivitelline membrane but not when swimming freely, even under capacitating conditions [12]. This unique patterned echidna sperm hyperactivation suggests specific ligands from the eggs might be required to activate the CatSper channel in monotreme sperm; CatSper activation mechanisms have diverged between monotremes and therian mammals. In addition, the egg-contact-dependent echidna sperm hyperactivation suggests that the significant motility change is particularly designed for egg penetration, rather than sperm migration. It is of note that additional layers of shells and mucoid around the oocyte are present in monotremes [55] (Figure 7). Therefore, sperm hyperactivation might have evolved as serving the function of penetrating egg barriers, which is conserved in all mammals, and later acquired additional roles for migration in the female reproductive tract in therian mammals.

## Figures and Tables

**Figure 1 cells-10-01047-f001:**
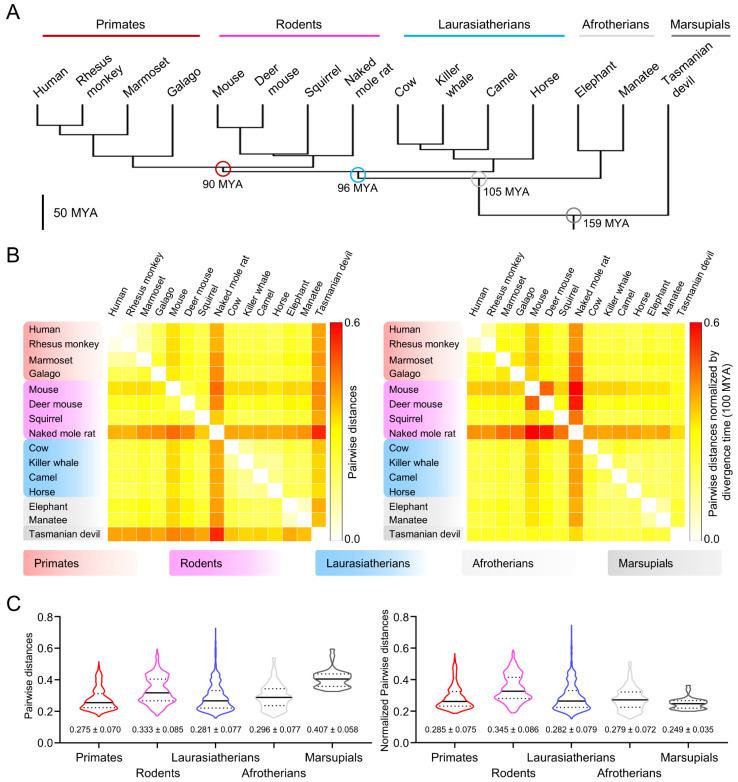
Amino acid sequence comparison of mammalian CatSper orthologs shows that CatSper channel components in marsupials evolved slowly. (**A**) A phylogenetic tree of placental mammals. Divergence times of the clades, primates (human, rhesus monkey, marmoset, and galago), rodents (mouse, deer mouse, squirrel, and naked mole rat), Laurasiatherians (cow, killer whale, camel, and horse), Afrotherians (elephant and manatee), and a marsupial (Tasmanian devil) are marked with circles. Million years ago, MYA. (**B**) Pairwise distance analysis of CatSper subunits between placental mammals. Concatenated protein sequences of CatSper subunits (CatSper1-2-3-4-β-γ-δ-ε-ζ-EFCAB9) were used to calculate the pairwise distances between species. Raw (left) and normalized (right) pairwise distances are represented with heatmaps. Values of the raw pairwise distances were normalized by divergence times between two species (100 MYA = 1). Pairwise distances over 0.6 were set to 0.6 (red). (**C**) Interclade comparison of the CatSper orthologs. Distributions of the raw (left) and normalized (right) pairwise distances between species in different clades were represented in violin plots. Solid and dotted lines in each plot indicate median and quartile values, respectively. Mean ± SD values for the raw and normalized pairwise distances were marked at the bottom of each column. See also Appendix A.

**Figure 2 cells-10-01047-f002:**
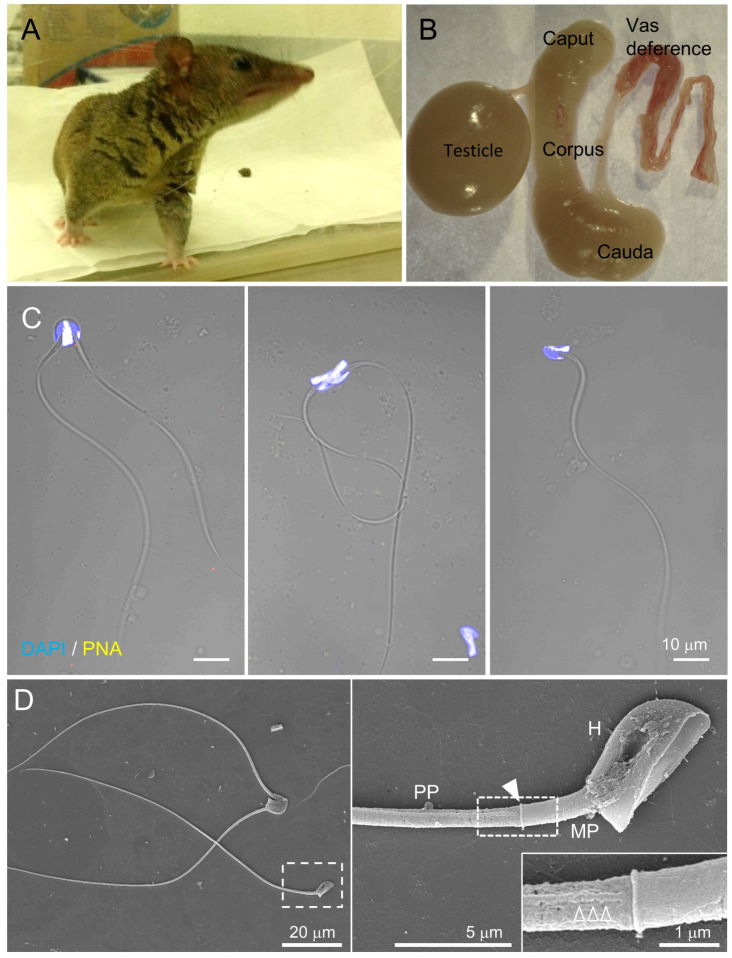
Gray short-tailed opossum sperm share subflagellar compartmentalization conserved in mammalian species. (**A**) An adult male gray short-tailed opossum (*Monodelphis domestica*). (**B**) A half side of the dissected male reproductive tract from an adult opossum. (**C**) Confocal images of the opossum epididymal sperm. Paired (left and middle) or single (right) sperm were observed in which PNA staining depicts the location of the acrosome on the anterior head. Hoechst was used for counterstaining. Fluorescence and corresponding DIC images were merged. (**D**) Scanning electron microscopy images of the opossum sperm. Shown are paired and single sperm (left). Sperm flagella is compartmentalized into midpiece (MP) and principal piece (PP), as shown in the enlarged inset (right). An arrowhead indicates the annulus, a junction between midpiece and principal piece. The magnified region in the right panel shows a longitudinal column (empty arrowheads). H, head. See also Appendix A.

**Figure 3 cells-10-01047-f003:**
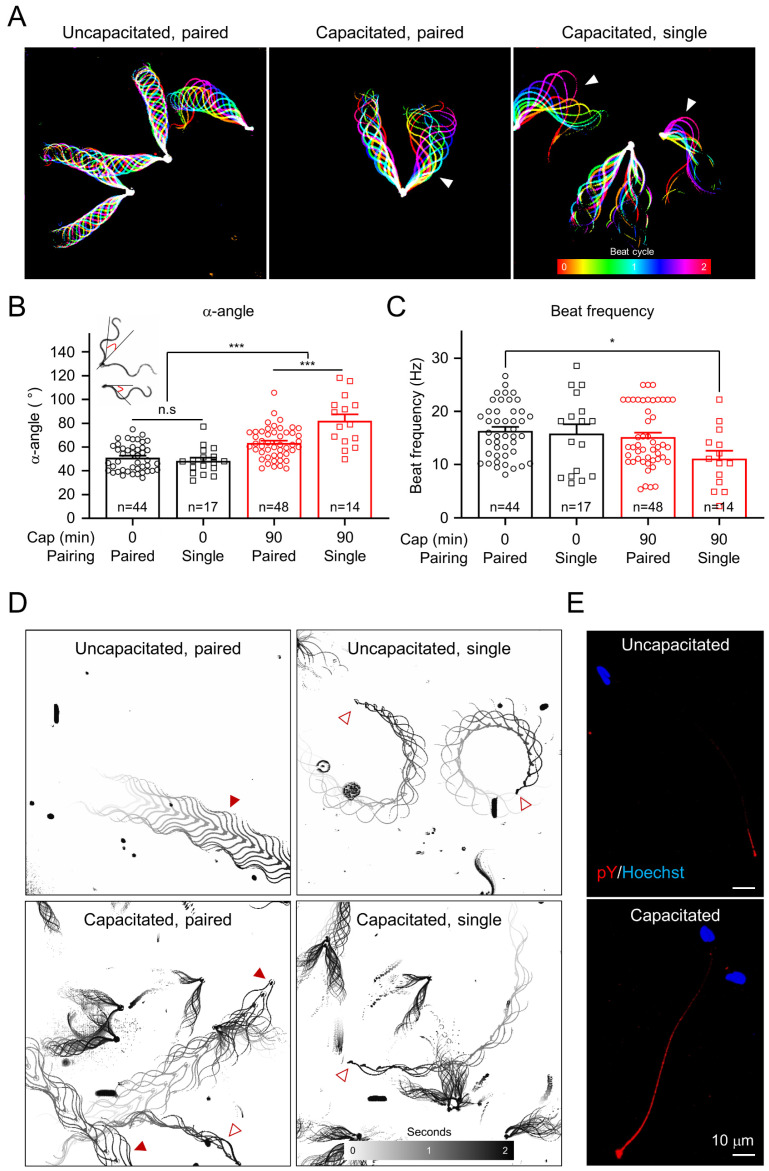
Capacitation induces changes in the motility pattern of gray short-tailed opossum sperm and tyrosine phosphorylation. (**A**) Flagellar waveform analysis. Sperm heads were tethered to the glass coverslips, and their tail beating was recorded from uncapacitated (left) and capacitated (middle and right) sperm. One flagellum of the paired sperm (middle) and an unpaired sperm flagellum (right) beat asymmetrically (arrowheads) after incubation under capacitating conditions. Flagellar waveforms from two-beat cycles were overlaid, and each frame was color-coded. (**B**,**C**) Quantitative comparisons of maximum angle of the primary curvature (**B**, α-angle) and beating frequency (**C**) of opossum sperm flagella. The α-angle (the red angles in sperm cartoon) and beat frequency were measured from paired (circles) and single (squares) sperm cells before (black) and after (red) inducing capacitation. α-angle of the uncapacitated sperm (0 min, paired, 51.0° ± 1.7°; unpaired, 48.3° ± 2.7°) increases after inducing capacitation for 90 min (90 min, paired, 63.7° ± 1.9°; single; 82.1° ± 5.5°). α-angle in single sperm is significantly larger than that in paired sperm after inducing capacitation. The beat frequencies of uncapacitated (0 min, paired, 16.3 ± 0.8 Hz; single, 15.8 ± 1.8 Hz) and capacitated (90 min, paired, 15.3 ± 0.9 Hz; single, 11.1 ± 1.5 Hz) opossum sperm were compared. n.s, non-significant, * *p* < 0.05, *** *p* < 0.001. Data are represented as mean ± SEM. (**D**) Swimming trajectories of the opossum sperm. Uncapacitated (top) and capacitated (bottom) opossum sperm were subject to free swimming under 0.5% methylcellulose, and swimming trajectories were drawn by overlaying time-lapse frames. Paired sperm swim in a straight line regardless of inducing capacitation (left). By contrast, unpaired single sperm swim in circles (right-top) with increased radius after capacitation is induced (right-bottom). Filled and empty arrowheads indicate free-swimming paired and single sperm, respectively. (**E**) Confocal images of the phosphotyrosine (pY)-immunostained opossum sperm. pY in opossum sperm were imaged before (top) and after (bottom), inducing capacitation. Hoechst was used for counterstaining. See also Appendix A.

**Figure 4 cells-10-01047-f004:**
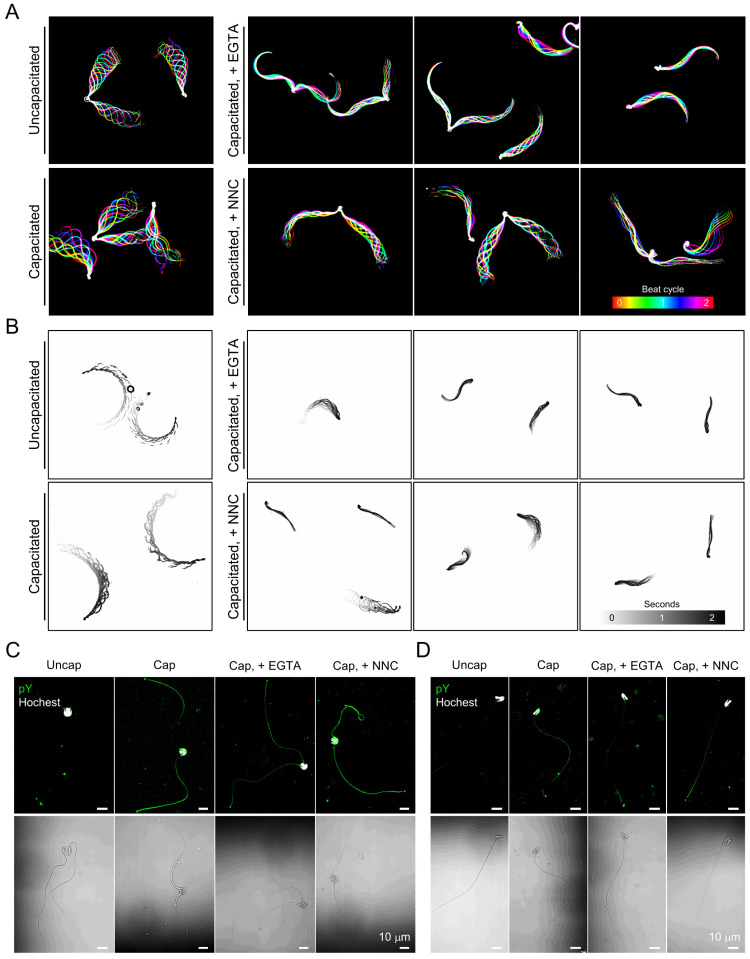
Inhibition of extracellular Ca^2+^ influx impairs hyperactivated motility in the capacitated opossum sperm. (**A**) Flagellar waveform analysis of sperm incubated under the capacitating condition with blocking Ca^2+^ entry. Opossum sperm were capacitated with EGTA (right-top) or CatSper channel inhibitor, NNC 55-0396 (NNC, right-bottom), and their flagellar movement was recorded. Inhibiting Ca^2+^ influx compromises opossum sperm to develop hyperactivated motility with increased flagellar amplitude. Flagellar movement during two beating cycles was overlaid and represented in color-coded images. (**B**) Swimming trajectory of the opossum sperm capacitated under impaired Ca^2+^ entry condition. Movements of free-swimming sperm were recorded for two seconds, and the swimming trajectory was visualized by overlaying time-laps frames. Sperm flagellar movement (**A**) and swimming trajectories (**B**) were recorded in H-HTF medium supplemented with 0.5% methylcellulose. (**C**,**D**) Confocal images of immunostained phosphotyrosine (pY) in paired (**C**) or single (**D**) opossum sperm capacitated with EGTA or NNC. Corresponding DIC images are shown below. Sperm heads were counter-stained with Hoechst. (**A**–**D**) 2.5 mM of EGTA and 10 mM of NNC were used. See also Appendix A.

**Figure 5 cells-10-01047-f005:**
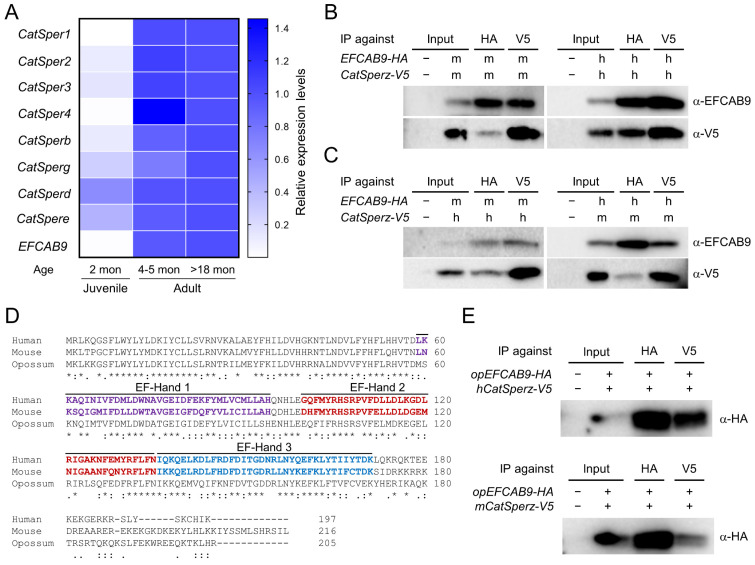
Gray short-tailed opossum shares molecular characteristics of CatSper subunits conserved among eutherians. (**A**) mRNA expression pattern of CatSper subunits in developing testes of the gray short-tailed opossum. Relative expression levels of CatSper subunits are represented in a heatmap from the qRT-PCR analysis. CatSper expression in the testis is higher in adults (4–5 months and over 18 months; *n* = 3 each) compared to juvenile males (2 months old; *n* = 2). Expression levels in over 18-month-old males were set to 1-fold. (**B**,**C**) Conserved interaction of eutherian EFCAB9 and CatSperζ. EFCAB9 and CatSperζ from the same (**B**) or different (**C**) species were transiently expressed in 293T cells and co-immunoprecipitated. m, mouse; h, human. (**D**) Amino acid sequence alignment of human, mouse, and opossum EFCAB9. Marked are the annotated three EF-Hands in human and mouse EFCAB9. (**E**) Opossum EFCAB9 can form a complex with CatSperζ orthologs from other eutherian mammals. Transiently expressed opossum EFCAB9 and human (top) or mouse (bottom) CatSperζ in heterologous system are co-immunoprecipitated. See also Appendix A.

**Figure 6 cells-10-01047-f006:**
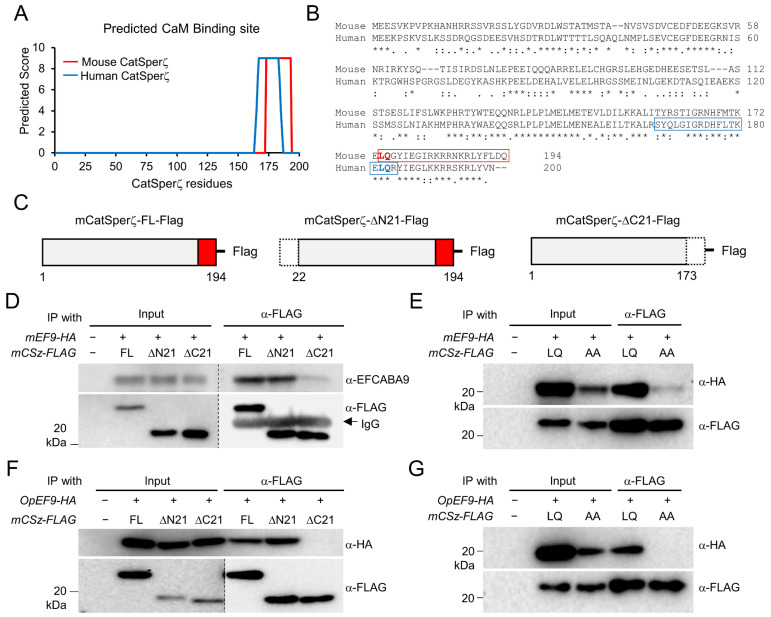
The IQ-like motif in the C-terminal CatSperζ is crucial for CatSperζ-EFCAB9 interaction conserved in placental mammals. (**A**,**B**) A putative calmodulin (CaM)-binding site conserved in mouse and human CatSperζ. (**A**) The predicted CaM-binding site in mouse (red) and human (blue) CatSperζ were scored and plotted. (**B**) Pair-wise sequence alignment of human and mouse CatSperζ proteins. The predicted CaM-binding sites are boxed. L174Q175 residues (bolded) in the CaM-binding regions are conserved in human and mouse CatSperζ. (**C**) Schematic diagrams of recombinant mouse CatSperζ (mCatSperζ) in full-length (right, FL; 1-194 aa) or 21-residue truncation at N-terminus (middle, ∆N21; 22-194 aa) or C-terminus (right, ∆C21; 1-173 aa). The candidate CaM-binding sites are red-boxed. (**D**–**G**) Co-immunoprecipitation of mouse CatSperz with defective IQ-like motif and mouse (**D**,**E**) or opossum EFCAB9 (**F**,**G**) in a heterologous system. Mouse CatSperz (*mCSz-FLAG*) with N- (DN21) or C- (DC21) termini truncation (**D**,**F**) or sequence mutation at IQ-like motif (L174A/Q175A, AA; **E**,**G**) were transiently expressed in 293T cells together with mouse (*mEF9-HA*) or opossum (*OpEF9-HA*) EFCAB9 in 293T cells and co-immunoprecipitated.

**Figure 7 cells-10-01047-f007:**
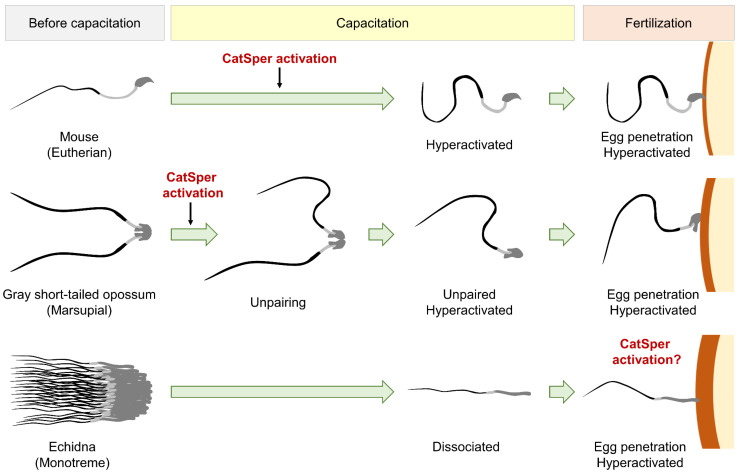
A schematic cartoon depicts sperm hyperactivation in each mammalian lineage. Development of sperm hyperactivation in three mammals, mouse (top), gray short-tailed opossum (middle), and echidna (bottom), which represent each lineage, eutherian, marsupials, and monotreme, respectively, are drawn. Mouse and gray short-tailed opossum sperm develop hyperactivated motility during capacitation, but echidna sperm requires egg contact after dissociation during capacitation to be hyperactivated. Sperm hyperactivation seems to be required to penetrate the egg barrier (right, orange layer) in all three lineages.

## Data Availability

No applicable.

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
