# Peer review of "Molecular Evolution of CatSper in Mammals and Function of Sperm Hyperactivation in Gray Short-Tailed Opossum"

_cells, 2021, doi:10.3390/cells10051047_

Round 1

Reviewer 1 Report

In this work, the authors investigated two different topics: 1) Evolutionary aspects of mammalian CatSper and 2) Sperm hyperactivation of marsupials using gray short-tailed opossum as the title indicates.

Related to the first topic, they demonstrated clearly that the C-terminus of CatSperζ have IQ-like motif that is responsible for the molecular interaction between CatSperζ and EFCAB9. A dramatic effect of two amino acids mutation of the IQ-like motif (from LQ to AA) on its interaction with EFCAB9 highlights their discovery. One weak point is that the genome information coding CatSperζ of gray short-tailed opossum is not available. However, the author demonstrated that EFCAB9 of gray short-tailed opossum interacts with mouse CatSperζ depending on the IQ-like motif, which suggest a conserved mechanism of interaction between CatSperζ and EFCAB9 in mammals. Another quite interesting finding in this work is high divergence of CatSper subunits in rodents compared to divergence found in other clades. Although physiological significance is not clear yet, this finding should be quite important to understand a certain particular function of CatSper in rodents.

The second part of study is characterization of sperm flagellar motility of gray short-tailed opossum. 

The second part of the study is characterization of sperm flagellar motility of gray short-tailed opossum. Several results support that hyperactivated sperm motility of this species dependents on CatSper. They also explored the Tyrosine phosphorylation during capacitation although its physiological relevance is currently unknown.

They also study the Tyrosine phosphorylation during capacitation although its physiological relevance is currently unknown.

Comments:

One of the import discoveries of this work is determination of the presence of IQ-like motif for the molecular interaction between CatSperζ and EFCAB9. Since the genome data of CatSperζ is not available in gray short-tailed opossum, it is impossible to perform biochemical experiments with CatSperζ of this species. Considering this situation, I recommend the authors to discuss about IQ-like motif of CatSperζ in other marsupials. At least, IQ-like motif (LQ) can be found in CatSperζ of Tasmanian devil (XP_031798511.1) although the IQ-like motif of Koala (XP_020823634.1) is LR instead of LQ. Probably, transcriptome of the testis of short-tailed opossum may help to identify CatSperζ of this species but a lot of time and cost will be required. It should be further study in the future.

I found a possible mistake in Fig. 1 B. Two boxes corresponding to the divergence between mouse and naked mole rat (Heterocephalus glaber) is white in right panel of Fig.1B. This while color (small difference) must be an error considering there is a significant difference of CatSper between two species (in Fig1B left panel and Fig. S1C). The color should be changed to real corresponding color.

Divergence of CatSper found among rodents is quite interesting. However, there is no description about this discovery in abstract. If further studies about this aspect are on going, the authors can report more detailed information in the subsequent paper. If not, the authors should mention this point in the abstract.

Minor error:

Line 69,

“distgantly” should be corrected to “distantly”

Author Response

#Reviewer 1

Point 1. One of the import discoveries of this work is determination of the presence of IQ-like motif for the molecular interaction between CatSperζ and EFCAB9. Since the genome data of CatSperζ is not available in gray short-tailed opossum, it is impossible to perform biochemical experiments with CatSperζ of this species. Considering this situation, I recommend the authors to discuss about IQ-like motif of CatSperζ in other marsupials. At least, IQ-like motif (LQ) can be found in CatSperζ of Tasmanian devil (XP_031798511.1) although the IQ-like motif of Koala (XP_020823634.1) is LR instead of LQ. Probably, transcriptome of the testis of short-tailed opossum may help to identify CatSperζ of this species but a lot of time and cost will be required. It should be further study in the future.

Response 1. We appreciate the reviewer’s suggestion. As suggested, we now have added the points and discussed further the conserved IQ-like motif-mediated interaction between CatSperz and EFCAB9 in mammals (Lines 550-552).

Point 2. I found a possible mistake in Fig. 1 B. Two boxes corresponding to the divergence between mouse and naked mole rat (Heterocephalus glaber) is white in right panel of Fig.1B. This while color (small difference) must be an error considering there is a significant difference of CatSper between two species (in Fig1B left panel and Fig. S1C). The color should be changed to real corresponding color.

Response 2. We apologize for this oversight. Indeed, the color code for the pairwise distance between mouse and naked mole rat was not set properly. We fixed the heatmap with corresponding color. Please see Figure 1B in revised manuscript (Page 6).

Point 3. Divergence of CatSper found among rodents is quite interesting. However, there is no description about this discovery in abstract. If further studies about this aspect are on going, the authors can report more detailed information in the subsequent paper. If not, the authors should mention this point in the abstract.

Response 3. Thank you for the reviewer’s suggestion. As shown in analysis, the divergence of CatSper proteins might be distinct in rodent species. This point is already briefly mentioned in Results. Indeed, we are now doing another study about this aspect but rather not include in abstract of this study. We appreciate your generous understanding.

Point 4. Line 69, “distgantly” should be corrected to “distantly”

Response 4. We appreciate the reviewer’s correction. We fixed the typo (Line 66 in revised manuscript).

Reviewer 2 Report

This is a high quality, elegant and sophisticated in terms of approaches study on opossum spermatozoa physiology including evolution. There is only on concern before it publication in a present form.
Give age and number of animals: gray short-tailed opossums (M. domestica), and wild type C57BL/6 mice. In addition passage and origin of cell line used in the study.

Author Response

#Reviewer 2

Point 1. Give age and number of animals: gray short-tailed opossums (M. domestica), and wild type C57BL/6 mice. In addition passage and origin of cell line used in the study.

Response 1. Thank you for the comments. We have added the information for age and number of animals (lines 104-106 and 108-109) and passage and origin of cell line (lines 182 and 184) in the revised manuscript.

Reviewer 3 Report

This is a very interesting study on the role of CatSper in sperm hyperactivity in mammals. The evolutionary aspect of this work deserves special attention and comparison not only with higher mammals, but also with the echidna. I believe that this kind of work deserves the highest praise.

I have only two points to make.

A minor remark is related to the first two paragraphs of the introduction: these are well-known facts, I believe that they are presented unreasonably long.

The second remark is fundamental. From the materials of the article, it follows that the authors used human tubular fluid for capacitation. However, data on the difference in the content of the human tubular fluid from the tubular fluid of the opossum and the mouse are not given. The regularities found by the authors in the hyperactivation of opossum and mouse spermatozoa may be associated with the same composition of the activating fluid. It cannot be ruled out that the use of a species-specific tubular fluid for capacitation may disprove the data obtained by the authors. I believe that the authors should provide data on the composition of the species-specific tubular fluid and discuss their results from this point of view. 

Author Response

#Reviewer 3

Point 1. A minor remark is related to the first two paragraphs of the introduction: these are well-known facts, I believe that they are presented unreasonably long.

Response 1. Thank you for the reviewer’s suggestion. In fact, we elaborated the introduction despite they are well-known facts with the purpose to invite the broad readership for this special issue in honor of Geoff A. Parker. Yet we see the reviewer’s point and trimmed down a small portion of the text.

Point 2. The second remark is fundamental. From the materials of the article, it follows that the authors used human tubular fluid for capacitation. However, data on the difference in the content of the human tubular fluid from the tubular fluid of the opossum and the mouse are not given. The regularities found by the authors in the hyperactivation of opossum and mouse spermatozoa may be associated with the same composition of the activating fluid. It cannot be ruled out that the use of a species-specific tubular fluid for capacitation may disprove the data obtained by the authors. I believe that the authors should provide data on the composition of the species-specific tubular fluid and discuss their results from this point of view

Response 2. We thank the reviewer for the constructive suggestion. The reviewer is correct that the fine composition of tubular fluid can be species-specific in mammals and that HTF used in our study potentially might have affected hyperactivation of opossum sperm. Our assumption is that major capacitating components such as nutrients, albumin, bicarbonate, and Ca2+ are essential factors to induce sperm capacitation across eutherian species as HTF (or similar medium containing these components) has been used to successfully capacitate sperm cells in multiple eutherian species in vitro (i.e., boar, bovine, equine sperm) while the percentage of hyperactivated sperm vary. Although the exact composition of oviductal fluids are not reported in marsupials, a previous study succeeded in vitro fertilization, followed by in vitro capacitation, in gray short-tailed opossum using modified MEM of which the chemical composition is also very similar to those of HTF (Moore and Taggart, 1993, J Reprod Fertil). Our using HTF to capacitate opossum sperm and our interpretation is based on this rationale. We discussed these points in lines 516-521 in the revised manuscript.